# Prolonged Hyperglycemia Causes Visual and Cognitive Deficits in *Danio rerio*

**DOI:** 10.3390/ijms231710167

**Published:** 2022-09-05

**Authors:** Elizabeth McCarthy, Jillian Dunn, Kaylee Augustine, Victoria P. Connaughton

**Affiliations:** Department of Biology, American University, Washington, DC 20016, USA

**Keywords:** zebrafish, three-chamber choice, blood sugar, optomotor response, T2DM

## Abstract

The present study induced prolonged hyperglycemia (a hallmark symptom of Type 2 diabetes [T2DM]) in *Danio rerio* (zebrafish) for eight or twelve weeks. The goal of this research was to study cognitive decline as well as vision loss in hyperglycemic zebrafish. Fish were submerged in glucose for eight or twelve weeks, after which they were assessed with both a cognitive assay (three-chamber choice) and a visual assay (optomotor response (OMR)). Zebrafish were also studied during recovery from hyperglycemia. Here, fish were removed from the hyperglycemic environment for 4 weeks after either 4 or 8 weeks in glucose, and cognition and vision was again assessed. The 8- and 12-week cognitive results revealed that water-treated fish showed evidence of learning while glucose- and mannitol-treated fish did not within the three-day testing period. OMR results identified an osmotic effect with glucose-treated fish having significantly fewer positive rotations than water-treated fish but comparable rotations to mannitol-treated fish. The 8- and 12-week recovery results showed that 4 weeks was not enough time to fully recovery from the hyperglycemic insult sustained.

## 1. Introduction

Diabetes mellitus (DM) is a serious metabolic disease that impacts 13% of the US adult population at 34.1 million, with over 90% of those cases being Type 2 (T2DM) [1]. T2DM is recognized by its decreased insulin sensitivity and high levels of blood glucose, i.e., hyperglycemia [2]. Hyperglycemia has been linked to both macrovascular (stroke and heart attacks) and microvascular complications (nephropathy, retinopathy, and peripheral neuropathy), as well as deficits in cognitive function, suggesting a brain pathology. Our work focuses on cognitive changes from prolonged hyperglycemia. 

Diabetic retinopathy (DR), a complication of diabetes, is the leading cause of blindness in working-age individuals and is responsible for 12% of all new blindness cases each year [3]. Within 20 years, 60–95% of DM cases progress to DR depending on the DM type and risk factor [3]. The main pathophysiological changes in DR are caused by hyperglycemia, which is considered a modifiable risk factor for DR [4,5].

Decreased cognitive function, especially weak episodic memory, cognitive inflexibility, and poor psychomotor performance, are also observed in DM patients, particularly T2DM [6]. There are many hypotheses as to the causes of cognitive decline including abnormal insulin signaling and oxidative stress [7]. Additionally, severe hyperglycemic states cause specific injury to the hippocampus and the cortex [8]. Breakdown of the blood–retina and blood–brain barriers, resulting in increased vascular permeability [9,10], is also induced by hyperglycemia.

In the past 20 years, zebrafish have been gaining popularity as a model organism not only to study DM but also to study other metabolic diseases. This is predominantly due to their small size (1–1.5 inches fully grown), high fecundity, prolific nature (spawning of 200 eggs per female), low cost, ease of maintenance, and transparency through the embryonic stages [2]. Additionally, zebrafish have organs and tissues that are similar in structure and function to those of humans, including the pancreas [11], and seem to display complex cognition comparable to mammals [12]. Of note, zebrafish retain the L1 region of the hippocampus, which plays a crucial role in memory consolidation in both zebrafish and mammals [13], and the mechanisms of glucose homeostasis are similar [11]. 

Since the initial publication [14], zebrafish have been developed as a model of DM in several ways. Type 1 DM (T1DM) can be induced through surgical removal of the pancreas, chemical-dependent ablation using streptozotocin or alloxan [11], and/or genetic ablation. The two main methods of induction for T2DM are glucose immersion [14,15,16,17], or a high fat diet [11]. Mutant and transgenic lines have also been created to study the effects of DM, the most common mutant being *pdx*1, a model of T2DM [11]. 

In this study, we induced hyperglycemia by submerging adult zebrafish in an alternating, increasing glucose solution (see [15]). At regular intervals during exposure (4, 8, and 12 weeks) we tested their cognitive performance, and at 8 and 12 weeks, we also tested the visual performance. Across the same time periods (i.e., 8 and 12 weeks), we also assessed the zebrafish’s ability to recover from hyperglycemic insult. Zebrafish are known to have regeneration capabilities, and they may be able to compensate for damages caused by prolonged hyperglycemia. For example, STZ ablated beta cells can be regenerated given time [16]. However, to our knowledge hyperglycemic insult and any subsequent recovery have not been compared at side-by-side timepoints. Thus, the ability to identify and track different complications over time and examine compensatory mechanisms are unique to this work. This study expands previous work by (1) extending hyperglycemia to 12 weeks, providing a comparison to 4-week and 8-week timepoints and (2) examining the effect of a recovery period (4 weeks of hyperglycemia followed by 4 weeks of recovery (8 weeks in total) and 8 weeks of hyperglycemia followed by 4 weeks of recovery (12 weeks total) on behavioral responses. 

## 2. Results

### 2.1. Blood Glucose Levels

After 8 weeks of glucose exposure, significant differences in blood glucose levels were identified across treatment groups, (F (2,17) = 5.064, *p* = 0.019). At this timepoint, glucose-treated fish had significantly higher mean blood glucose levels than both water-treated and mannitol-treated fish (*p* = 0.037, *p* = 0.048 respectively) (Figure 1A).

After 12 weeks of treatment, glucose exposure still had a significant effect (F (2,29) = 6.298, *p* = 0.005) on blood sugar levels. Here, glucose-treated fish again had significantly higher mean blood glucose levels than both water-treated and mannitol-treated fish (*p* = 0.010, *p* = 0.011, respectively) (Figure 1B).

In both of our recovery conditions (8-week washout and 12-week washout), after 4 weeks removed from the hyperglycemic environment, glucose-treated fish no longer had significantly different blood glucose levels from the controls (Figure 1C,D).

### 2.2. Three-Chamber Choice: Hyperglycemia; Full Exposure

#### 2.2.1. Discrimination Ratios and Trendlines—8 Weeks of Hyperglycemia

A significant overall effect of treatment on discrimination ratios was observed after 8 weeks of treatment (F (8,278) = 4.219, *p* < 0.001; Figure 2, single asterisk). However, the day was the only variably significant (F (2,278) = 13.462, *p* < 0.001), whereas treatment and the interaction between treatment and day were not significant. More specifically, discrimination ratios on reversal day 1 were similar, and near 0.4, for all three treatment groups. On day 2, the ratio for the water-treated group was lower than the ratios for glucose- and mannitol-treated groups. By day 3, ratios in all groups were increased, with the lowest score in the glucose-treatment group. 

Further, mannitol- and water-treated fish had significant trendlines (*p* = 0.007, and *p* < 0.001, respectively) and a significant difference between the mean discrimination ratios on day 1 and day 3 (*p* = 0.006, *p* < 0.001, respectively) (Figure 2, double asterisks). Glucose-treated fish, however, had neither a significant trendline nor a significant difference between reversal days 1 and 3. This suggests that fish in the mannitol and water (control) groups were successfully learning the task over the 3 days, so that by day 3, these fish more consistently swam to the side of the chamber with the shoal reward. In contrast, fish in the hyperglycemic environment were not learning/had difficulty learning the new location of the shoal within the three-day period or remembering where the reward had been previously. Overall, glucose-treated fish displayed a 50% chance of selecting the correct side by day 3, compared to a >60% chance observed in control fish.

#### 2.2.2. Percent Correct, Time Comparison: 4-Week vs. 8-Week Trends

To determine the prolonged effect of hyperglycemia, we compared the mean percent correct score from 4-weeks reversal day 3 (4.3) of hyperglycemia with the mean percent correct scores collected on all 3 reversal days after 8 weeks of hyperglycemia (day 1, day 2, day 3) (Figure 3). Comparison of 4-week vs. 8-week response revealed that fish in the glucose- (Figure 3B) and mannitol- (Figure 3C) treatment groups displayed significantly fewer correct scores on all 8-week days (day 1 *p* < 0.001, day 2 *p* < 0.001, day 3 *p* ≤ 0.001). Responses from these fish did show slight improvement in the percentage of correct scores from day 1 to day 3, though the scores remained below the 4-week day 3 value. Fish in the water-treatment group (Figure 3A) also had a significantly lower percent correct score on 8-week day 1 compared to 4.3 (*p* = 0.05); however, the scores of these fish showed improvement with each 8-week day, so that the 8-week day 3 values were not significantly different from the 4.3 score. These data suggest that after 8 weeks of treatment, fish in all three groups initially (day 1) went to the side of the chamber, where they remembered the reward that was located previously (4.3). However, by day 3, the water-treated fish were the only ones that were successfully able to find the shoal reward and learn its new location within the three-day time constraint.

#### 2.2.3. Discrimination Ratios and Trendlines—12 Weeks of Hyperglycemia

A significant overall effect treatment on discrimination ratios was also observed at the 12-week timepoint (F (8,138) = 1.945, *p* = 0.05, Figure 4). However, in this case, treatment was the only variable to have an overall significant effect (F (2,138) = 4.904, *p* = 0.009), whereas the day and interaction between the treatment and day were not significant. Discrimination ratios calculated for the 12-week timepoint (where the shoal was placed back on the side with the black background) revealed overall better responses on day 1, particularly in the mannitol- and glucose-treatment groups (Figure 4, single asterisk). Ratios calculated for water-treated fish were consistently the lowest over all 3 days. On day 3, glucose and mannitol displayed discrimination ratios of ~0.8, much higher than the 0.6 water control value. This, however, is not indicative of new learning/improvement; rather, it is indicative of memory from where the shoal was placed at 4 weeks and were placed once again. Glucose-treated fish never successfully learned where the shoal was at 8 weeks and therefore had no inhibitory mechanisms stopping them from choosing the correct side at 12 weeks.

At 12 weeks, there were no significant trendlines or differences between reversal days 1 and 3 for any treatment (Figure 4). However, the mean discrimination ratios were >50% for all three days regardless of treatment, though the calculated ratios for water-treated fish dipped just below 50% on R2 (49%).

#### 2.2.4. Percent Correct, Time Comparison: 8-Week vs. 12-Week Trends

As above, we compared the number of correct responses observed on the third day of 8-week reversal testing (8.3), with the numbers of correct responses observed on all three 12-week reversal days (day 1, day 2, day 3) (Figure 5).

In this analysis, hyperglycemic glucose-treated fish (Figure 5B) again displayed a significant difference in their responses on all three 12-week days vs. 8.3. However, now, the glucose-treated fish had significantly more correct responses (day 1 *p* = 0.02; day 2 *p* = 0.006; day 3 *p* < 0.001). Fish in the mannitol-treatment group (Figure 5C) significantly increased their response only at 12-week day 3 compared to 8.3 (*p* = 0.003) and fish in the water-treatment group (Figure 5A) had significantly fewer correct responses on 12-week day 2 (*p* = 0.0023).

These results show that glucose-treated fish did not learn to reverse the task after 8 weeks of exposure, allowing them to accurately choose the correct response on all 12-week days. This suggests that these glucose-treated fish show evidence of memory but no new learning or improvement in the task as their scores do not increase over time; the percent correct scores for all three days at 12 weeks (~75%) are roughly the same as they were on 4.3 (Figure 2). Water- and mannitol-treated controls, in contrast, have responses on day 1 that are at/below the 8-week scores, indicating that they chose the side where the reward was previously located. Both fish in both control groups show increasing scores with each day, indicating that they were able to find the shoal and learn its new location.

### 2.3. Three-Chamber Choice: Hyperglycemia; Recovery

#### 2.3.1. Discrimination Ratios and Trendlines—8-Week Washout (4 Weeks of Hyperglycemia Followed by 4 Weeks of Recovery)

We performed the same analysis as above to assess whether removal from treatment (i.e., return to normoglycemia) affected behavior.

No significant effects due to treatment, reversal day, or treatment * reversal day were identified in 8-week washout fish. At this timepoint, all discrimination ratios on day 1 were similar and near 0.5 (Figure 6). On the following two days, the discrimination ratios for all three treatment groups remained clustered together, though mannitol- and water-treated fish had the highest discrimination ratios on day 2, whereas mannitol- and glucose-treated fish had the overall highest discrimination ratios on day 3.

Similarly, trendline analysis identified no significant differences across reversal days (Figure 6). Overall, these results suggest that the hyperglycemic fish were able to ‘recover’ when assessed 4 weeks after removal from treatment, as there was no difference between their responses and the responses of controls.

#### 2.3.2. Percent Correct, Time Comparison: 4-Week Exposure vs. 8-Week Washout Trends

Comparison over time (as in Section 2.2.2) showed that fish in the glucose-treatment group (Figure 7B) had significantly lower percent correct scores on all 8-week reversal days (day 1 *p* < 0.001; day 2 *p* = 0.001; day 3 *p* = 0.05) compared to the scores at 4.3. This suggests that the glucose-treated fish remembered the task well after 4 weeks and that three days was not enough time for them to fully learn a new task. Further, the consistent significance seen here was also observed in fish that were hyperglycemic for 8 weeks, with no recovery period (Figure 2), albeit to a stronger degree, suggesting that while recovery may be initiated, it has not been completed. There was no significant difference found between days 2 and 3 of testing, further substantiating the idea that recovery was not fully reached.

The mean percent correct scores of mannitol-treated fish (Figure 7C) show a similar trend, with 8-week day 1 being significantly lower than scores at 4.3 (*p* = 0.002), indicating that fish learned the new location of the shoal reward and suggesting that the results observed in the glucose-treatment group are not due to a general osmotic effect. In contrast, water-treated fish (Figure 7A) had responses similar to values at 4.3 on all 8-week reversal days except day 2 (*p* = 0.004), indicating that the fish were able to quickly learn the new location of the shoal reward.

Overall, this analysis indicates that mannitol-treated fish, and to a lesser extent, glucose-treated fish, given a recovery period were able to learn the task, based on the growing scores over the three-day period. This trend was not observed in the results from the 8-week continuous hyperglycemia experiment.

#### 2.3.3. Discrimination Ratios and Trendlines—12-Week Washout (8 Weeks of Hyperglycemia Followed by 4 Weeks of Recovery)

We next assessed whether 8 weeks of hyperglycemia followed by 4 weeks of normoglycemia (recovery) altered the behavioral responses. For this experiment, a significant overall effect (F (8,82) = 2.952, *p* = 0.006) due to treatment (F (2,82) = 7.051, *p* = 0.001) was observed. The day and the interaction between the treatment and day did not significantly affect responses.

Specifically, on day 1, water-treated fish had the highest discrimination ratio (~0.6), while the ratio for glucose-treated fish was just under 0.5, and the ratio for mannitol treated fish was the lowest (~0.3) (Figure 8, asterisk). On day 2, the discrimination ratios for all the treatment groups increased, with water-treated fish still having the highest score and mannitol-treated fish having the lowest. On day 3, glucose- and water-treated fish both had lower discrimination ratios than on day 2; however, the ratio for mannitol-treated fish increased. By the third day, all fish had similar discrimination ratios of ~0.6.

Neither glucose- nor mannitol-treated fish had significant trendlines or a significant difference between day 1 and day 3. Water-treated fish had a significant deviation to their trendline (*p* = 0.05); however, there was no significant difference across any of three days (Figure 8).

At this stage of testing, there was no significant difference in blood glucose levels between any groups. These data look to the differences due to recovery between treatment groups. While no learning appears to take place in the glucose-treated group, this group shows some recovery in glucose-specific mechanisms due to its similar pattern to water-treated fish and differences from mannitol-treated fish.

#### 2.3.4. Percent Correct, Time Comparison: 8-Week Exposure vs. 12-Week Washout Trends

The percent of correct responses in glucose-treated fish that were hyperglycemic for 8 weeks (8.3) were next compared to the percent correct responses of 12-week fish given a recovery period. In this analysis (Figure 9), percent correct scores of fish in glucose treatment were not significantly different on any 12-week reversal day (day 1, day 2, day 3) when compared to scores on 8.3 (Figure 9B). In contrast, fish in the water-treated group (Figure 9A) had significantly higher scores on 12-week day 2 than on 8.3 (*p* = 0.021), suggestive of learning, though scores were significantly reduced on day 3. Fish in the mannitol-treated group (Figure 9C) had fewer correct responses on day 2 compared to 8.3 (*p* = 0.021), but by day 3, there was no longer a significant difference, again suggestive of learning.

### 2.4. OMR after 8 and 12 Weeks of Hyperglycemia

After 8 weeks of hyperglycemia, significant differences in optomotor responses were observed across treatment groups (F (2,60) = 6.795, *p* = 0.002; Figure 10A). Glucose-treated fish had significantly fewer completed rotations on average than water-treated fish, but not mannitol-treated fish (*p* = 0.002; Figure 10A).

After 12 weeks of hyperglycemia, glucose exposure also had a significant overall effect (F (2,29) = 3.954, *p* = 0.03) on optomotor responses. Glucose-treated fish again made significantly fewer completed rotations on average than water-treated fish, but not mannitol-treated fish (*p* = 0.033; Figure 10B).

## 3. Discussion

In the present study, we show that 8 weeks of hyperglycemia induced cognitive and visual deficits overall. At 12-weeks, cognitive deficits were still observed, though to a lesser extent than at 8 weeks, and visual deficits were still present. In particular, 8-week washout (recovery) results showed slight learning in glucose-treated fish after 4 weeks of recovery, where no learning was evident in the 8-week full exposure (no recovery) results. This indicates that while a 4-week recovery period is not enough time to fully change responses in the three-chamber choice task, even though blood sugar values returned to normglycemic levels, the results indicate the initiation of recovery.

At 12 weeks, a 4-week recovery period showed similar discrimination ratios for water-treated and glucose-treated fish, which suggests recovery. However, trendline comparison showed that water-treated fish had better learning patterns than glucose-treated fish over the three days, which suggests that, once again, 4 weeks was not enough time to recover. These data are summarized in Figure 11.

Overall, we found that prolonged hyperglycemia for either 8 weeks or 12 weeks in the zebrafish model reduced learning in the three-chamber choice task. If the fish were allowed to return to normglycemic conditions, different trends were noted, suggesting some recovery. Our results also show that the duration of hyperglycemic insult prior to normglycemia is an important variable for the recovery process.

While we do note that average discrimination ratios measured ~60%, in many of our analyses, this was not a surprising finding. Previous research [18] separated the fish as “high” and “low” performers based on their performance in the three-chamber choice task (high-performing fish scored 75% or better on discrimination day 3, whereas low-performing fish scored below 75% on discrimination day 3) and analyzed the data separately for these two groups. Their findings are consistent with our results, with both high- and low-performing water-treated fish averaging between 60 and 80% by day 3 of testing. This identifies the importance of looking at both discrimination ratios (trendline analysis) and percent correct scores (timeline comparison) to accurately determine the effect of hyperglycemia.

The ability to extend hyperglycemia research is important for two reasons: (1) some complications are reported to take longer to develop than others, and (2) our previous work suggests that zebrafish may be able to compensate in continued hyperglycemic conditions. Compensation can also be seen in STZ-injected zebrafish, which are able to regenerate their beta cells after a single injection [16]. We aimed to address these questions here by extending the duration of hyperglycemic insult and by examining the impact of a washout/recovery period.

### 3.1. Hyperglycemia and Cognitive Behaviors

Prolonged hyperglycemia is linked to cognitive decline both in both rodents [9] and zebrafish [2]. Further, different animal models of DM show cognitive deficits at different timepoints. For instance, STZ mice show cognitive deficits 6 weeks after induction, while BB/Wor rats and NOD mice (both a spontaneous model of T1DM) do not show deficits until after 8 weeks [19]. In zebrafish, these cognitive deficits can be brought on as quickly as 2 weeks in a high-fat diet model of T2DM [12,20].

Our previous work showed that glucose-treated fish have worse cognitive performance than mannitol-treated fish and water-treated fish after 4 weeks of hyperglycemia [18] (in review). However, that study also identified osmotic differences in discrimination ratios after 8 weeks of hyperglycemia. Comparison of percent correct scores at 8 weeks shows the same trends for glucose- and mannitol-treated fish, consistent with this previous work. Similarly, we did not find a difference in discrimination ratios due to treatment on any reversal day at 8 weeks. Nonetheless, trendline analysis clearly separated responses of glucose-treated vs. control fish at this timepoint.

Overall, these results indicate that at the 8 week timepoint glucose-treated fish were not able to successfully learn the reversed location of the shoal reward. This was evident when assessing both percent correct scores and discrimination ratio trendline analysis. In contrast, water-treated fish did successfully learn the reversed shoal location. Mannitol-treated fish, however, showed evidence of learning only in the trendline analysis; percent correct scores during reversal suggest that these fish did not correctly find the shoal reward.

After 12 weeks of hyperglycemia, we expected to see a worsening of responses, given that hyperglycemic insult was extended. However, at this timepoint, the fish displayed increased percent correct scores, indicating that they were easily able to identify the reversed shoal location. Further consideration of these data suggest that this better-than-expected response occurred because glucose-treated fish remembered where the shoal was placed from the 4-week testing point. The shoal reward was on the same side of the chamber at 4 weeks and 12 weeks. Thus, hyperglycemic zebrafish learned the task well during initial training and continued to remember it, despite subsequent reversal testing. In contrast, water- and mannitol-treated fish both showed evidence of learning and were clearly able to identify the reversed location of the shoal reward.

Overall, these findings suggest that the effects observed after 8 weeks of hyperglycemia are complex, influenced by osmotic load, and clearly different from the earlier 4-week timepoint. Further, continued hyperglycemic insult to 12 weeks did not cause greater deficits in learning. Rather, it revealed that the initial learning of the task was maintained in glucose-treated fish, resulting in high numbers of individual percent correct scores, even though discrimination ratios from each trial were not different.

### 3.2. Recovery from Hyperglycemic Insult

When zebrafish are given STZ injections to ablate their beta cells, recovery can happen very quickly after drug removal, with fish returning to normoglycemic values after 2 weeks without the use of insulin injections [16,21]. Zebrafish can similarly recover after pancreas removal, with increased division of existing β-cells compensating for the loss [21]. So, while the zebrafish might be returning to normoglycemic levels in our washout experiments, we were interested in the lasting implications of hyperglycemic damage. After 4 weeks of glucose exposure followed by 4 weeks of recovery, the trendline analysis showed no significant difference between groups, suggesting recovery. However, when the percent correct scores were analyzed, the glucose-treated group did not show robust evidence of learning where the water- and mannitol-treated groups did. In fact, the percentage of correct scores from glucose-treated fish in the 8-week washout experiment are similar to the percent of correct scores in the 8-week full exposure (no washout) protocol. This indicates that while the fish may be recovering, four weeks may be too short a duration to say that glucose-treated fish have returned to baseline.

In the 12-week washout experiments, fish were exposed to 8 weeks of glucose exposure, with 4 weeks of recovery. In this group, treatment-specific differences in discrimination ratios were observed on reversal days 1 and 2; however, trendline analysis and percent correct scores were not different. In fact, on all three 12-week reversal days, glucose-treated fish responded similar to 8-week reversal day 3 responses, suggesting that the fish did not learn the new location of the shoal. In contrast, water-treated fish easily identified the location of the shoal (based on their high percent correct scores), and mannitol-treated fish learned the location over the three reversal days. This again suggests that while 4 weeks is enough time for blood sugar levels to return to normoglycemic levels, it is not enough time to reverse the cognitive decline seen after 4 and 8 weeks of prolonged hyperglycemia.

### 3.3. Optomotor Responses with Prolonged Hyperglycemia

We previously showed that after 4 weeks and 8 weeks of hyperglycemia, zebrafish adults have increased positive OMRs [18]. This finding is opposite to what we observed here, wherein glucose-treated fish had the lowest OMR scores.

The OMR behavioral assay provides evidence of a fully functioning visual system and can be used as a measure of visual acuity. When zebrafish are suffering from visual deficits, they will swim in random patterns and will not follow in the direction of the stimulus [17]. Past research has shown that hyperglycemia causes harm to the retina [17,22] and is correlated with thinner retinal layers [14,16] altered retinal vasculature [23], and increased glycation of proteins within the eye [24]. At both 8 and 12 weeks, hyperglycemic fish had significantly reduced positive OMRs compared to water-treated fish. We do not know why this result is different from our previous work; however, it does point to osmotic effects as glucose-treated fish did not have significantly different scores to mannitol-treated fish at either timepoint.

### 3.4. Comparison Hyperglycemic Effects on Vision and Cognition

It is important to understand whether our results represent glucose-specific or osmotic effects. If the results were to be osmotic, we would expect glucose and mannitol results to look similar, but different from water-treated fish. If the results were glucose-specific, we would expect mannitol- and water-treated fish to look similar and glucose-treated fish results to appear different. While the observed optomotor responses seem to be driven by general osmotic differences, the cognitive effects seem to point to a more glucose-specific impact. Timepoint comparison among 12-week exposure, 8-week washout, and 12-week washout data all identified a glucose-specific effect, as did the trendline analysis for the 8-week exposure and 12-week washout groups (summarized in Table 1). However, trendline analysis of 12-week exposure data and timepoint comparison of 8-week exposure data both point to osmotic effects, with glucose- and mannitol-treated fish showing similar responses. This differing of pathways also seems to be dependent, in part, on the type of analysis run. This could be due to the difference in discrimination ratios vs. percent correct scores, or it could be to do with the analysis itself. Both analyses are powerful tools to look at the data, and neither should be discounted; however, it is interesting that they do not always show the same result. Further, it is important to continue to study the link between hyperglycemia and cognition to discover specific mechanisms of general osmotic vs. glucose-specific pathways that impact cognition and vision.

In conclusion, we show an association between hyperglycemia and learning at 8 weeks of prolonged hyperglycemia and to a lesser extent at 12 weeks in the zebrafish model. Additionally, our washout/recovery experiments suggest that 4 weeks was not enough time for the zebrafish to fully recovery from the hyperglycemic insult, even though they returned to normoglycemic levels quickly, and that the duration of hyperglycemia impacts recovery. Lastly, we also found that vision (OMR) was impaired in glucose-treated fish. Further research should be carried out to look at the molecular components leading to these results such as the breakdown of the blood–brain barrier and the blood–retina barrier, as well as looking into the causes of these breakdowns such as the increase in inflammatory cytokines and the increase in oxidative stress.

## 4. Materials and Methods

### 4.1. Animals

Adult wild-type zebrafish (*Danio rerio*) aged 4–12 months were obtained from a commercial supplier (Live Aquaria; Rhinelander, WI, USA) or bred in-house, and were kept in the Zebrafish Ecotoxicity, Neuropharmacology, and Vision (ZENV) laboratory at American University. Fish were held in an Aquatic Habitat (AHAB, Pentair, Apopka, FL, USA) at ~28 °C and on a 14–10 h light–dark cycle until needed for experiments. Zebrafish were fed daily using commercial flakes (TetraMinTM, Blacksburg, VA, USA) and live brine shrimp (*Artemia*, Connecticut Valley, Southampton, MA, USA). All fish were randomly chosen for participation, with both males and females used in all experiments, and randomly separated into the different treatment groups. Upon the completion of the experiment, animals were anesthetized in 0.02% tricaine, decapitated, and tissue was collected for later investigation. All experimental procedures were approved by the Institutional Animal Care and Use Committee (IACUC) at American University (protocol #19-02 renewed #22-08).

### 4.2. Induction of Hyperglycemia and General Experimental Design

To induce hyperglycemia, zebrafish were placed in 4 L tanks maintained at 28–29 °C. Fish were fed daily before transfers, at which time the pH and temperature were recorded. Glucose was used as our experimental substance against two controls: (1) water, a stress control; and (2) mannitol, an osmotic control. Mannitol is structurally similar to glucose and is known to have osmotic effects when administered. We include mannitol exposure as a treatment group to determine whether hyperglycemia (glucose exposure) is causing glucose-specific effects or general osmotic changes.

Hyperglycemia was induced using a stepwise alternate immersion protocol [15,25]. In brief, zebrafish were transferred from a tank containing one of the three solutions to a tank containing only water every 24 h. This is meant to mimic the increase and decrease of glucose in diabetics. More specifically, for the first two weeks (i.e., weeks 1 and 2), fish were maintained in a 1% solution (40 g); for the following two weeks (weeks 3 and 4), fish were maintained in a 2% (80 g) solution. If the experiment continued for either 8 weeks or 12 weeks, the fish were maintained in a 3% (120 g) solution for 4 weeks beginning at week 4, and in a 4% solution (160 g) for 4 weeks starting at week 8.

Two “washout” experiments were conducted to examine recovery from hyperglycemic insult. These experiments lasted either a total of 8 weeks (4 weeks in a hyperglycemic environment and 4 weeks in recovery) or 12 weeks (8 weeks in a hyperglycemic environment and 4 weeks in recovery). When fish were moved to recovery, they were all placed in designated stock tanks within the AHAB system and maintained in similar light, food, and temperature conditions as they had been in previously. Fish were directly moved from 2% solution (8-week washout) or 3% solution (12-week washout) and put into system water (0% solution) (Figure 12).

“Full” and “Washout” experiments were performed side by side not only to evaluate recovery from hyperglycemic insult but also to mimic the rise and fall of glycemic control seen in human diabetes. It is important to study the lasting implications of poor glycemic control due to the longevity of DM.

Fish were tested in the three-chamber choice task every 4 weeks (Figure 12), after the onset of treatment, to assess their cognitive abilities [26]. At the 8-week and 12-week timepoints, visual discrimination was also tested using an optomotor response (OMR). All assays were tested in fresh system water (0% solution), after which they were placed back into their respective treatment groups. Once both behavioral tests were complete, a subset of fish was anesthetized in a 0.02% tricaine solution, weighed (wet weight, Sartorius microbalance), and decapitated (using a sharp razor blade slicing anterior to the pectoral fin) so that the blood sugar levels could be measured (cardiac blood, FreeStyle Lite Blood Glucose meter; Abbott Diabetes Care Inc., Alameda, CA, USA). Brain and retinal tissue were flash frozen or fixed in 4% paraformaldehyde for later molecular analysis.

### 4.3. Three-Chamber Choice Assay

The behavioral chamber and experimental set up can be found in [26] and are only briefly described here. The chamber was made from a 40 L aquarium (50 × 30 × 30 cm^3^). The aquarium was then split into three chambers: a center chamber (10 × 30 × 30 cm^3^) and two side chambers (20 × 30 × 30 cm^3^). Opaque PVC sheets acted as separators for the chambers, and the aquariums were filled to 25 cm from the top of the aquariums, or roughly 30 L. Aquariums were filled with system water and heated to 28 °C. Two tanks were run simultaneously, and scoring was performing by two trained observers.

#### 4.3.1. Acclimation

Acclimation included one day of group acclimation followed by two days of individual acclimation. During group acclimation, up to six zebrafish were tested simultaneously. For acclimation, both side chambers had neutral (beige-colored) backgrounds and a clear container holding four conspecific zebrafish to act as the shoal reward [26]. The walls of the center chamber were not colored. Fish were initially placed into the center chamber for 2 min. The doors separating the center chamber from the side chambers were then opened for 30 min, giving the fish time to explore the entire chamber and identify the shoal reward.

For the subsequent two days, individual fish were placed in the center chamber with the doors closed and allowed to acclimate for 2 min. Then, the doors were opened, giving the fish access to the side chambers. A “choice” was made when the body of the fish had completely passed over a chamber door, either left or right. To successfully complete individual acclimation, a fish had to make 10 choices within 30 min, regardless of which side. If the fish was unable to complete the 10 choices within 30 min on the second day of individual acclimation, it was excluded from the experiment.

#### 4.3.2. Discrimination

Discrimination began the day after acclimation. At this time, the backgrounds of the two side chambers were changed from a neutral (beige) to a biased design with white fabric on the left side and black fabric on the right. The shoal was now only placed on the side covered with white fabric. Fish were again placed into the center chamber and given 2 min to acclimate. Then, both chamber doors were opened, giving each fish access to both sides. Each fish performed the task eight times (trials). For each fish, each trial was scored as a correct, incorrect, or marked choice. If the zebrafish chose the correct side (i.e., the side that contained the shoal reward) then the chamber door was closed to allow 1 min to interact with the ‘reward’. This trial was scored as “C” for correct and was considered a “rewarded” trial. If the fish chose the chamber without the shoal, they were immediately moved back to the center chamber and the trial was marked “I” for incorrect and was considered a “non-rewarded” trial. Lastly, if the fish remained in the center chamber for 2 min after the doors had been opened, the fish would be moved to the correct chamber for 1 min and would be coded as a “M” for marked and would be considered a “forced rewarded” trial.

#### 4.3.3. Reversal

After 4, 8, and 12 weeks of exposure, reversal learning was assessed for each fish, using the same chamber as above. Starting at 4 weeks, this meant that the shoal was now placed on the opposite-colored side of the tank (reversed). At 4 weeks, the shoal was placed on the black side; at 8 weeks, the shoal was placed on the white side; and at 12 weeks, the shoal was again placed on the black side.

As above, correct/incorrect/marked choice was determined for all trials. Reversal trials were completed every other day so that the fish were coming out of treatment solution (i.e., glucose, mannitol, or water) when they were being tested. This was performed to ensure that the blood sugar levels were elevated (i.e., the fish were hyperglycemic) during the testing period. We previously identified normoglycemic levels in fish after 24 h of water exposure [14].

### 4.4. OMR

Optomotor responses (OMRs) were recorded by placing two fish in a cylindrical dish that was placed on top of a flat-panel computer monitor. Fish were allowed to acclimate for 30 s before stimulus presentation. The stimulus was a rotating black and white radial grating projected beneath the fish [27]. The stimulus first rotated clockwise for 30 s followed by 30 s of counterclockwise rotation, repeated twice. A gray screen was projected for 30 s between each stimulus to act as both a control and a rest for the fish. Responses to the stimulus were recorded from above using a Canon video camera (VIXIA HFR700, 32× optical zoom, 57× advanced zoom, HD; Canon, Tokyo, Japan). Videos were scored by watching the recording playback and counting the number of complete revolutions made by each fish during each stimulus presentation. A positive OMR occurred when the fish swam in the direction of the stimulus.

### 4.5. Statistical Analysis

#### 4.5.1. Blood Glucose Levels

All statistical tests were run on SPSS v28. Blood glucose measurements were assessed using a one-way ANOVA at the 8-week and 12-week timepoints. Treatment was the main effect and significance was evaluated at α = 0.05.

#### 4.5.2. Three-Chamber Choice

On each day of testing, eight trials were run per fish. For each fish, the total number of correct trials was divided by total trials to give each fish a discrimination ratio. We also calculated the percentage of correct scores for all fish within a given treatment. Statistical differences in discrimination ratios and percent correct were determined separately using a one-way ANOVA followed by a Bonferroni post hoc test.

To determine if there was an effect of glucose exposure on learning, we performed a trendline analysis for each treatment group across the 3 days of testing at 8 weeks and 12 weeks.

Finally, we used *t*-tests to assess differences among the final day of the previous 4-week exposure to the three days of testing at 8 weeks or the final day of 8-week exposure compared to the three days of 12-week testing. This analysis was performed to assess memory over the 4-week period between testing and learning over the three-day period of testing.

#### 4.5.3. Three Chamber Choice: Discrimination Ratios vs. Percent Correct

Analysis for three-chamber choice data was performed in two ways: by fish and by trial. Analysis by trial was performed by calculating a percent correct score. In this analysis, the total number of correct scores by all fish in the selected trial are summed up and divided by the total number of scores in that trial, which varies depending on how many fish are part of the group. Percent correct scores were used to compare behavior during each reversal timepoint (i.e., reversal at 4-week day 3 with reversal at 8 weeks).

Discrimination ratios [26] were calculated by dividing the number of correct trials by the total number of trials (8) per fish, which was then averaged out over the group. While similar to the percent correct in that both are fractions with 1.00 = 100% and 0.5 = 50%, the discrimination ratios provided a better analysis of individual fish responses. Discrimination ratios were used to compare responses across reversal days (i.e., 8-week reversal day 1 vs. day 2 vs. day 3).

We also performed an analysis of the trendlines for individual treatment groups, specifically focusing on the differences in discrimination ratios between reversal day 1 and day 3. If there was a significant difference between day 1 and day 3 for a treatment group, giving rise to a significant trendline, we considered that this showed learning over the 3-day period.

#### 4.5.4. OMR

A one-way ANOVA was performed with treatment as the independent variable and mean rotations while the stimulus was used as the dependent variable to determine whether there was a difference across groups. If a post hoc test was needed, unless otherwise stated, the Bonferroni post hoc analysis was used.

## Figures and Tables

**Figure 1 ijms-23-10167-f001:**
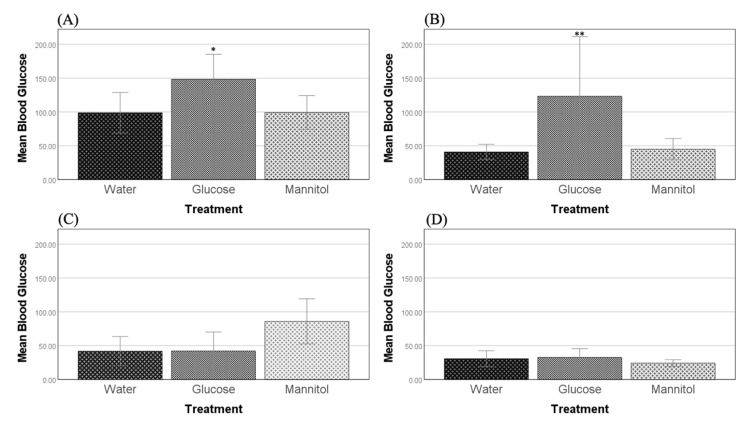
Blood glucose levels. Blood glucose levels were measured after (**A**) 8 weeks and (**B**) 12 weeks of hyperglycemia. At both timepoints, glucose-treated fish had significantly higher blood sugar levels than either water- or mannitol-treated controls (one-way ANOVA, 8 weeks, *p* = 0.037, *p* = 0.048, * respectively; 12 weeks, *p* = 0.010, *p* = 0.011, ** respectively). Values are means +/−95% CI. *n* = 23 (8 weeks); *n* = 32 (12 weeks). Four weeks of recovery after either four weeks of hyperglycemia (**C**) or eight weeks of hyperglycemia (**D**) showed that glucose-treated fish were no longer significantly different from either of the controls. Values are means +/−95% CI. *n* = 18 (8-week washout); *n* = 30 (12-week washout).

**Figure 2 ijms-23-10167-f002:**
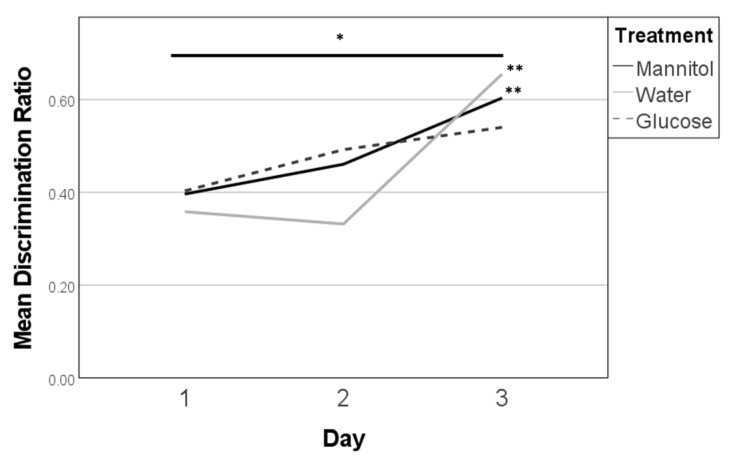
Discrimination ratios after 8 weeks of prolonged hyperglycemia. Discrimination ratios were calculated on all three 8-week reversal days (day 1, day 2, day 3) for responses of fish in each treatment group and analyzed on each day using one-way ANOVA. A significant difference due to reversal day (single asterisk) was noted (F (2,278) = 13.462, *p* < 0.001). Subsequent trendline analysis identified significant trends in responses for mannitol and water-treated fish (double asterisk) characterized by increased scores from day 1 to day 3 (*p* = 0.007, and *p* < 0.001, respectively). Significant trend lines suggest that the fish learned the new location of the shoal reward. A significant trend line was not found for glucose-treated fish (dashed line).

**Figure 3 ijms-23-10167-f003:**
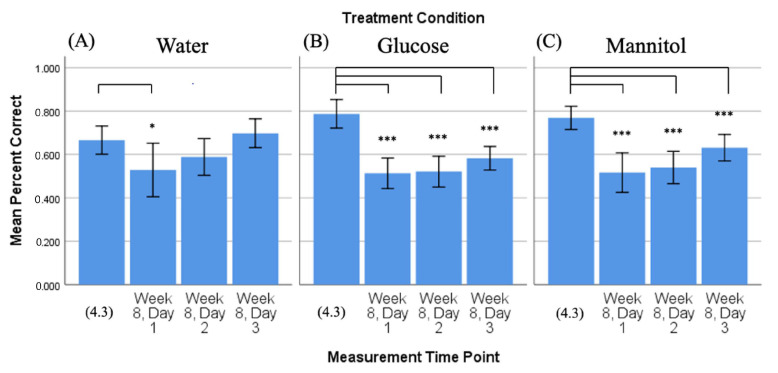
Comparison of 4-week vs. 8-week reversal. Mean percent correct scores (+/−95% CI) calculated on 4-week reversal day 3 (4.3) were compared to scores calculated on all three 8-week reversal days for (**A**) water-treated, (**B**) glucose-treated, and (**C**) mannitol-treated fish using individual *t*-tests. Water-treated fish showed an improvement in scores over the three 8-week reversal days, indicating they were able to learn the new location of the shoal reward. In contrast, glucose- and mannitol-treated fish had significantly lower numbers of correct scores on all three 8-week reversal days (day 1 *p* < 0.001, day 2 *p* < 0.001, day 3 *p* ≤ 0.001), suggesting these fish were unable to learn the new location of the shoal reward. * and ***: Asterisks denote significant differences.

**Figure 4 ijms-23-10167-f004:**
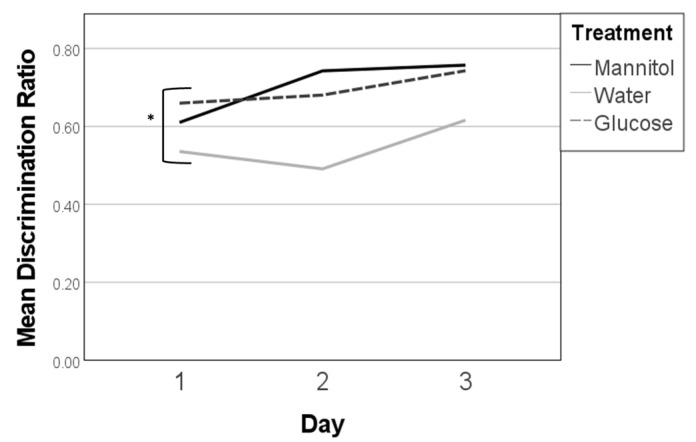
Discrimination ratios after 12 weeks of prolonged hyperglycemia. Discrimination ratios were calculated on all three 12-week reversal days (day 1, day 2, day 3) for responses of fish in each treatment group. A significant difference due to treatment was noted on day 1 only (one-way ANOVA, F (2,138) = 4.904, *p* = 0.009 *), wherein responses from glucose- and mannitol-treated fish were larger than water-treated controls. Subsequent trendline analysis did not identify any significant trends in responses across all three days of the shoal reward. A significant trend line was not found for glucose-treated fish (dashed line).

**Figure 5 ijms-23-10167-f005:**
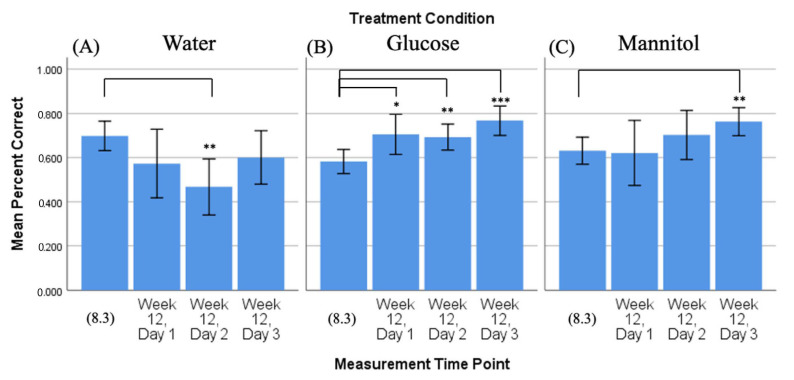
Comparison of 8-week vs. 12-week reversal. Mean percent correct scores (+/−95% CI) calculated on 8-week reversal day 3 (8.3) were compared to scores calculated on all three 12-week reversal days for (**A**) water-treated, (**B**) glucose-treated, and (**C**) mannitol-treated fish using individual *t*-tests. Water-treated fish showed a reduction in correct scores on day 2 (panel (**A**), **). Glucose-treated fish showed significantly more correct scores on all three 12-week reversal days (day 1 *p* = 0.02; day 2 *p* = 0.006; day 3 *p* < 0.001), indicating that these fish were able to find the location of the shoal reward. *,**,*** Asterisks in (**B**) denote significant differences, where the number of asterisks denotes level of significance. Mannitol-treated fish showed significantly increased scores by day 3 (*p* = 0.003, (panel (**C**), **)).

**Figure 6 ijms-23-10167-f006:**
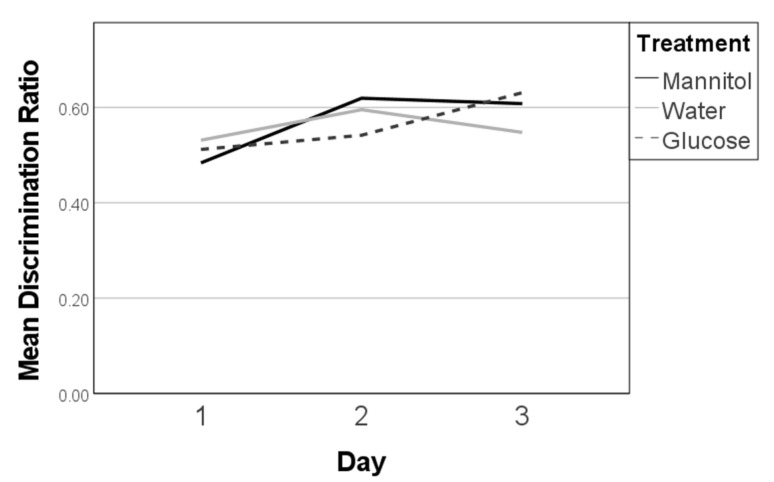
Discrimination ratios in the 8-week washout experiments. Discrimination ratios were calculated on all three 8-week reversal days (day 1, day 2, day 3). In this experiment, fish were exposed to hyperglycemic conditions for 4 weeks, followed by 4 weeks in normglycemic (system water) conditions. No significant differences in discrimination ratios were observed on any reversal day (*p* > 0.05; one-way ANOVA). Subsequent trendline analysis also failed to identify any significant differences.

**Figure 7 ijms-23-10167-f007:**
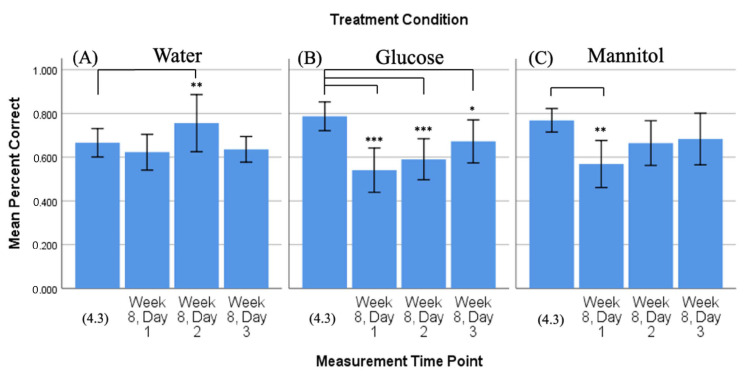
Comparison of 4-week vs. 8-week washout reversal. Mean percent correct scores (+/−95% CI) calculated on 4-week reversal day 3 (4.3) were compared to scores calculated on all three 8-week reversal days for (**A**) water-treated, (**B**) glucose-treated, and (**C**) mannitol-treated fish. In these experiments, 8-week fish were maintained in hyperglycemic conditions for 4 weeks, followed by a 4-week removal from treatment. Water-treated fish displayed a significant increase in percent correct scores on day 2 only (*p* = 0.004, panel (**A**), **); scores on other days were not different from 4.3. Glucose-treated fish had a significantly lower number of correct scores on all three 8-week reversal days (day 1 *p* < 0.001; day 2 *p* = 0.001; day 3 *p* = 0.05; individual *t*-tests performed on each day. *,*** Asterisks in (**B**) denote significant differences with the number of asterisks reflecting level of significance). Mannitol-treated fish had significantly lower correct scores on day 1 only (*p* = 0.002, **).

**Figure 8 ijms-23-10167-f008:**
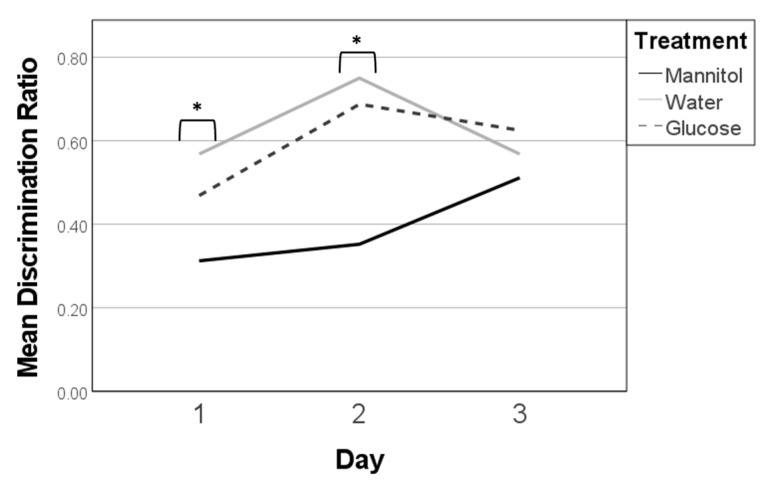
Discrimination ratios in the 12-week washout experiments. Discrimination ratios were calculated on all three 12-week reversal days (day 1, day 2, day 3). In this experiment, fish were exposed to hyperglycemic conditions for 8 weeks, followed by 4 weeks in normglycemic (system water) conditions. Significant differences in discrimination ratios were observed on days 1 and 2, when the responses of mannitol treated fish were lower than responses of fish in the glucose- or water-treatment groups (one-way ANOVA, F (2,82) = 7.051, *p* = 0.001, *). However, subsequent trendline analysis did not identify any significant differences across reversal days.

**Figure 9 ijms-23-10167-f009:**
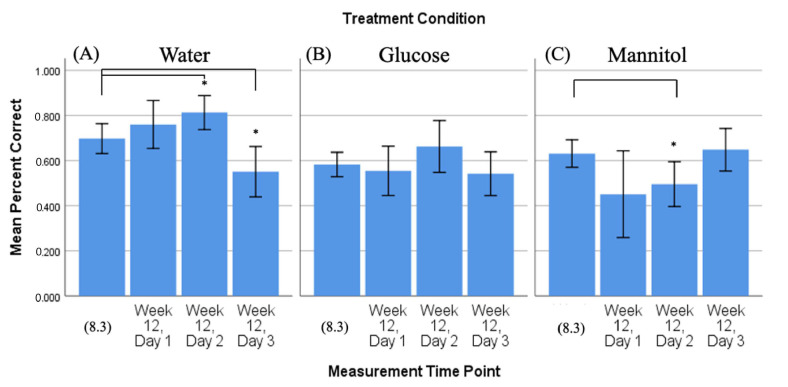
Comparison of 8-week vs. 12-week washout reversal. Mean percent correct scores (+/−95% CI) calculated on 8-week reversal day 3 (8.3) were compared (individual *t*-tests) to scores calculated on all three 12-week reversal days for (**A**) water-treated, (**B**) glucose-treated, and (**C**) mannitol-treated fish. At the 12-week timepoint, the fish had been exposed to hyperglycemic conditions for 8-weeks, followed by 4 weeks of recovery. Water-treated fish showed an increase in correct scores at day 2 (*p* = 0.021), but a decrease on reversal day 3. The percent correct scores in glucose-treated fish were not different across all three reversal days or compared to 8.3. Mannitol-treated fish also had responses similar to 8.3, though the percentage of correct scores was reduced on reversal day 2 (*p* = 0.021). Asterisks denote significant differences.

**Figure 10 ijms-23-10167-f010:**
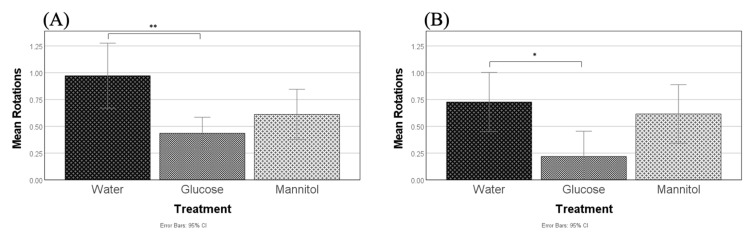
Optomotor responses. Optomotor responses (OMRs) were measured after (**A**) 8 weeks and (**B**) 12 weeks of hyperglycemia. At both timepoints, one-way ANOVA identified reduced positive OMRs in glucose-treated fish compared to water-treated controls (8-weeks, *p* = 0.002, **; 12-weeks, *p* = 0.033, *). Values presented are means +/−95% CI.

**Figure 11 ijms-23-10167-f011:**
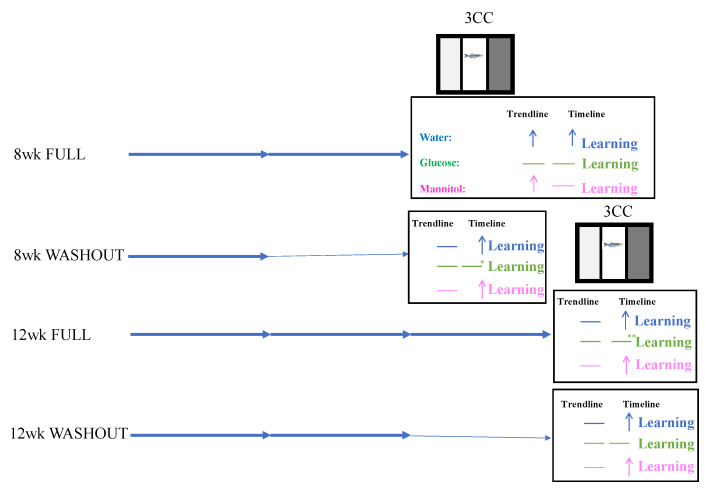
Overview of cognitive and visual performance. For each experiment: 8 wk Full = 8 weeks of hyperglycemia; 8 wk WO = 8-week washout experiment with fish exposed to glucose for 4-weeks followed by a 4-week recovery period; 12 wk Full = 12 weeks of hyperglycemia; 12 wk WO = 12-week washout experiment with fish exposed to glucose for 8-weeks followed by a 4-week recover period. Here we can see that water-treated fish displayed learning in over 50% of the analyses (denoted by the arrow), whereas mannitol displayed learning in exactly 50% of the analyses. Glucose-treated fish displayed learning in 0% of the analyses with two exceptions. At 8 wk washout denoted by the single asterisk, glucose-treated fish showed evidence of slight learning although they did not pass the threshold, and at 12-weeks full denoted by the double asterisk glucose-treated fish showed evidence of memory, but no new learning.

**Figure 12 ijms-23-10167-f012:**
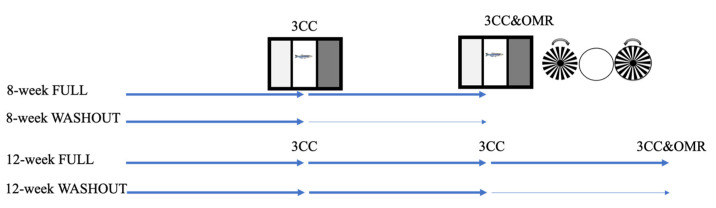
Experimental design. Using the alternating, increasing immersion protocol, hyperglycemia was induced in adult zebrafish for either 8 weeks (8-week FULL) or 12 weeks (12-week FULL). To assess the effect of recovery, additional experiments were performed in which the fish were hyperglycemic for 4 weeks, followed by 4 weeks removed from treatment (8-week WASHOUT) or hyperglycemic for 8 weeks, followed by 4 weeks removed from treatment (12-week WASHOUT). Each arrow in the above figure represents 4 weeks; darker arrows reflect treatment exposure, and thin arrows represent recovery. At each 4-week timepoint, cognitive behavior was assessed using the three-chamber choice task (3CC). At 8 weeks and 12 weeks, optomotor responses (OMRs) were also recorded.

**Table 1 ijms-23-10167-t001:** Glucose-specific vs. osmotic pathways. This table summarizes whether a given behavior appears to be due to a general osmotic effect or a glucose-specific effect. 8 wk Full = 8 weeks of hyperglycemia; 12 wk Full = 12 weeks of hyperglycemia; 8 wk WO = 8-week washout experiment with fish exposed to glucose for 4 weeks followed by a 4-week recovery period; 12 wk WO = 12-week washout experiment with fish exposed to glucose for 8 weeks followed by a 4-week recover period.

	Trendline Analysis	Timepoint Comparison	OMR
8 wk	Glucose specific	Osmotic	Osmotic
12 wk	Osmotic	Glucose specific	Osmotic
8 wk WO	Osmotic	Glucose specific	NA
12 wk WO	Glucose specific	Glucose specific	NA

## Data Availability

Data available from the authors upon request.

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
