# Peer review of "Prolonged Hyperglycemia Causes Visual and Cognitive Deficits in Danio rerio"

_ijms, 2022, doi:10.3390/ijms231710167_

Round 1
Reviewer 2 Report
Scientists preferred utilizing rodents for preclinical studies of diabetes mellitus but in nowadays Danio rerio is use as a preferable laboratory animal. Scientists found it interesting to study diabetes mellitus in Danio rerio as an animal model. Diabetes mellitus, whose main cause is hyperglycemia, affects the central nervous system, leading to neurodegenerative diseases.
Diabetes mellitus have been induced by a variety of methods in Danio rerio (zebrafish). Zebrafish pancreas development is very homologous to mammals, such as mice. The signaling mechanisms and way the pancreas functions are very similar. The pancreas has an endocrine compartment, which contains a variety of cells. Beta-cells, that produce insulin, are an example of those such cells. This structure of the pancreas, along with the glucose homeostasis system, are helpful in studying diseases, such as diabetes, that are related to the pancreas.
The majority of work done surrounding knowledge on glucose homeostasis has come from work on zebrafish transferred to humans.
The presented study confirms that a high-glycemic environment induced hyperglycemia in the zebrafishes, which are affecting their cognitive abilities as well as vision.
Danio rerio were also studied during recovery from hyperglycemia and results identified an osmotic effect with glucose-treated fish.
The study shows the important consequences of diabetes mellitus at the eye level and at the central nervous system level. The preclinical study is elaborated, it is well written and documented. I recommend it for publication.
Author Response
Thank you for the positive comments and endorsement.
Reviewer 3 Report
In this manuscript, McCarthy et al. examined the effect of prolonged hyperglycermia on cognition and vision using zebrafish as a model. They concluded that hyperglycemia impaired learning ability in zebrafish and vision, which cannot be reversed even after 4 weeks washout period. Overall this manuscript is difficult to read. Some of the data did not support their conclusion.
1. It seems that hyperglycemia does not impairs fish's ability to learn, instead glucose treatment increase it compared to water treatment (Fig. 3). Furthermore, compared to 8-week treatment (Fig. 2), 12-week treatment increases mean discrimination ratio (Fig. 3).
2. In Fig. 7, it seems the difference is mainly due to mannitol treatment. The difference between water and glucose is very minimal. It is unclear how water and mannitol treatment have so dramatic difference considering they both have similar glucose levels. Therefore, it suggested that glucose may not be the major contributor to the mean discrimination ratio.
3. In Fig. 8, is the difference between day 1, 2 and 3 significant to support the claim that glucose- and mannitol-treated fish given a recovery period were able to learn the task, based on the growing scores over the three-day period.
Minor issues:
1. Line 96, 3.2 should be 2.2
2. Line 97, 3.2.1 should be 2.2.1
3. Line 116, 3.2.2 should be 2.2.2
4. Line 202, day 1 should be day 3.
5. Fig. 2, no error bar was shown although it was claimed error bar: 95% CI.
6. The section of 3.2.1 can be moved to material and methods.
7. It is better to include statistical analysis in each figure legend.
Reviewer 4 Report
Dear Editor and Authors,
Regarding the present manuscript, the following should be mentioned:
1. The first five references from Introduction chapter should be replaced with newer evidence form the medical literature.
2. The term "type 2" is prefered instead of "type II" when referring to diabetes classification.
3. Material and Methods section could be placed after Introduction chapter.
4. Most figures (1,2,3,6,7,10) presented in the Results chapter are quite difficult to read and should be modified for better understanding.
5. The Results chapter contain a lot of informations, covering 9 pages. It could be reduced to the most important findings of the present study.
6. No new results should be placed in the Discussion chapter.
5. the subchapter 3.2 should be placed in the Methods chapter and also data regarding statistical analysis mentioned in other subchapters.
Round 2
Reviewer 3 Report
The authors claimed that they have added statistical analysis in figure legends. However, I did not see any statement about statistical analysis in Figure legends. They need to clear state which statistical analysis was performed in figure legends.
Author Response
Thank you for your comment, we apologize for any confusion beforehand. We believe that we have now successfully added a statement of statistical analysis to all of our figures.